# Genetic Elucidation for Response of Flowering Time to Ambient Temperatures in Asian Rice Cultivars

**DOI:** 10.3390/ijms22031024

**Published:** 2021-01-20

**Authors:** Kiyosumi Hori, Daisuke Saisho, Kazufumi Nagata, Yasunori Nonoue, Yukiko Uehara-Yamaguchi, Asaka Kanatani, Koka Shu, Takashi Hirayama, Jun-ichi Yonemaru, Shuichi Fukuoka, Keiichi Mochida

**Affiliations:** 1National Agriculture and Food Research Organization, Institute of Crop Science, Tsukuba, Ibaraki 305-8518, Japan; kzfmngt@affrc.go.jp (K.N.); nonouey221@affrc.go.jp (Y.N.); zhengf@yahoo.co.jp (K.S.); yonemaru@affrc.go.jp (J.-i.Y.); fukusan@affrc.go.jp (S.F.); 2Institute of Plant Science and Resources, Okayama University, Kurashiki 710-0046, Japan; saisho@rib.okayama-u.ac.jp (D.S.); hira-t@rib.okayama-u.ac.jp (T.H.); 3RIKEN Center for Sustainable Resource Science, Yokohama 230-0045, Japan; yukiko.uehara@riken.jp (Y.U.-Y.); asaka.kanatani@riken.jp (A.K.); 4Kihara Institute for Biological Research, Yokohama City University, Yokohama 244-0813, Japan

**Keywords:** rice, flowering time, ambient temperature fluctuation, chromosome segment substitution line (CSSL), quantitative trait locus (QTL)

## Abstract

Climate resilience of crops is critical for global food security. Understanding the genetic basis of plant responses to ambient environmental changes is key to developing resilient crops. To detect genetic factors that set flowering time according to seasonal temperature conditions, we evaluated differences of flowering time over years by using chromosome segment substitution lines (CSSLs) derived from *japonica* rice cultivars “Koshihikari” × “Khao Nam Jen”, each with different robustness of flowering time to environmental fluctuations. The difference of flowering times in 9 years’ field tests was large in “Khao Nam Jen” (36.7 days) but small in “Koshihikari” (9.9 days). Part of this difference was explained by two QTLs. A CSSL with a “Khao Nam Jen” segment on chromosome 11 showed 28.0 days’ difference; this QTL would encode a novel flowering-time gene. Another CSSL with a segment from “Khao Nam Jen” in the region around *Hd16* on chromosome 3 showed 23.4 days” difference. A near-isogenic line (NIL) for *Hd16* showed 21.6 days’ difference, suggesting *Hd16* as a candidate for this QTL. RNA-seq analysis showed differential expression of several flowering-time genes between early and late flowering seasons. Low-temperature treatment at panicle initiation stage significantly delayed flowering in the CSSL and NIL compared with “Koshihikari”. Our results unravel the molecular control of flowering time under ambient temperature fluctuations.

## 1. Introduction

Global warming is likely to reach 1.5 °C between 2030 and 2052 if it continues at the current rate [1]. Cultivated rice (*Oryza sativa* L.) originated from a wild rice species (*Oryza rufipogon* Griff.), which grows mainly in the tropics [2,3]. Rice is inherently adaptable to hot environments. However, novel rice cultivars with resistance to very hot environments are required for regions where extreme warming is expected [1]. Climate change is causing large fluctuations in temperature, solar irradiation, precipitation and soil moisture, especially in the tropics [4]. At the high-latitude limits of rice cultivation; however, low temperatures at the seedling, panicle initiation and maturation stages severely decrease grain yield [5,6]. There is now an increasing demand for new cultivars that are adaptable to both effects of climate change. Therefore, to develop climate-change resilient crops, it is necessary to elucidate the genetic basis of plant response to ambient environmental changes.

Daylength is a primary environmental factor that determines maturation (harvest) time through induction or suppression of flowering and consequently adaptability to each growing area in plant species [7,8]. Rice is a short-day plant: flowering is promoted under short daylength. Most genes for flowering cloned so far are involved in response to daylength and photoperiod sensitivity in rice [9,10,11,12,13]. They are assigned to two major independent gene regulatory pathways, the *OsGI*–*Hd1*–*Hd3a* pathway and the *Ghd7*–*Ehd1*–*Hd3a*/*RFT1* pathway. Both pathways are regulated by light perception, daylength and the circadian clock. These previous molecular genetic studies show the importance of response to daylength in the determination of rice flowering time.

Temperature is another important ambient environmental factor that determines flowering time. Vernalization (low-temperature) treatment promotes flowering in several cereal and horticultural species such as wheat, barley and *Brassica* vegetables [14,15]. In Arabidopsis, higher temperature accelerates plant growth and development through signaling from *PhyB*, *PIF*s, *DELLA* and the evening complex, which includes *ELF3* [16,17,18,19,20]. Previous studies revealed that rice cultivars adapted to higher latitudes showed photoperiod insensitivity and high-temperature response [21,22,23]. Four flowering-time QTLs responsive to temperature were detected in a mapping population derived from crosses between *indica* and *japonica* cultivars [24]. In addition, other previous studies investigated relationships between rice flowering times under different temperature conditions and expression levels of previously isolated genes in rice mutant lines [25,26]. They showed that expression of *Ehd1*, *Hd3a* and *RFT1* was repressed under long daylength and low temperatures. The data suggest that temperature response is a decisive factor in determining flowering time in rice. However, despite their importance in relation to breeding programs, the molecular mechanisms involved in temperature response remain largely unknown. It is necessary to elucidate the genetic mechanisms underlying the determination of flowering time in response to ambient temperature in rice.

Here, we evaluated differences of flowering time over 9 years (2011 to 2019) in Asian rice cultivars and chromosome segment substitution lines (CSSLs) derived from crosses between *japonica* rice cultivars “Koshihikari” and “Khao Nam Jen”. “Khao Nam Jen” had a large difference. Among the 40 CSSLs, two lines with a “Khao Nam Jen” segment on chromosomes (Chrs.) 11 and 3 (the latter including the *Hd16* region) had over 15 days’ difference. The QTL on Chr. 11 would be a novel flowering-time gene, because no allelic differences have been found in genes previously isolated in this region. Low-temperature treatment at the panicle initiation stage delayed flowering time in both lines. RNA-seq analysis revealed that expression of the flowering-time genes *Ehd1*, *Hd3a* and *RFT1* was shifted later in the flowering season in the CSSL than in “Koshihikari”. Through these analyses, we detected novel genetic factors involved in the control of flowering time under different ambient temperatures during the rice-growing season.

## 2. Results

### 2.1. Natural Variations in Flowering Time Among Years

We found wide variations in flowering time (days-to-heading, DTH) of 18 Asian rice cultivars during 2011 to 2019 (Figure 1A, Appendix A). For example, DTH varied from 87.6 ± 1.1 (extremely early) in “Qui Zhao Zong” to 201.4 ± 2.3 (extremely late) in “Khao Nam Jen” in 2011 and from 88.4 ± 2.3 (extremely early) in “Qui Zhao Zong” to 215.0 ± 0.5 (extremely late) in “Nona Bokra” in 2019. Among the 18 cultivars, “Khao Nam Jen” and “Bei Khe” had a large difference of flowering time: 36.7 and 23.2 days, respectively. We calculated coefficients of variations (CVs) to evaluate dispersions of flowering time in the 18 cultivars for nine years. “Khao Nam Jen”, “Bei Khe”, “Muha” and “Tupa 121-3” had large CVs: 5.6, 5.4, 5.7 and 5.2. On the other hand, “Bleiyo” had a small difference of flowering time: 3.2 days. “Bleiyo”, “LAC23” and “Basilanon” had small CVs: 1.5, 2.7 and 2.8, respectively. Daylength is strictly stable, while ambient temperature fluctuates widely among years (Figure 1B,C). Flowering time in “Bleiyo”, “LAC23′ and “Basilanon” was strongly controlled by daylength, while that in “Khao Nam Jen”, “Bei Khe”, “Muha” and “Tupa 121-3” was easily affected by ambient temperature during the cropping period. Therefore, we selected “Khao Nam Jen”, with the largest flowering time difference, to genetically dissect the molecular basis for flowering time fluctuations among years through analysis of progeny derived through backcrossing to “Koshihikari”, which had a moderate difference of flowering time (9.9 days; CV = 3.0).

### 2.2. Flowering Time Fluctuation in CSSLs of “Koshihikari” and “Khao Nam Jen”

To identify genetic factors related to the flowering time fluctuation, we evaluated flowering time in a set of CSSLs derived from crosses between “Koshihikari” and “Khao Nam Jen” from 2011 to 2019. The CSSLs, developed in a previous study, comprise 40 lines at the BC_4_F_6_ generation (SL2801–SL2840; Appendix A; [27]). Each CSSL has a substituted segment of a target chromosomal region from “Khao Nam Jen” in the genetic background of “Koshihikari”. Flowering times of most CSSLs were similar to those of “Koshihikari” in each year (Figure 2). However, several CSSLs had large differences in flowering time from “Koshihikari”. Line SL2833, with a “Khao Nam Jen” segment on the short arm of Chr. 11, had a flowering time difference of 28.0 days and CV = 9.6 (Figure 2 and Figure 3; Appendix A). SL2812, with a “Khao Nam Jen” segment on the long arm of Chr. 3, had a flowering time difference of 23.4 days and CV = 6.5. Most other CSSLs had a flowering time difference of ~15 days and CV ≤ 5, similar to those of “Koshihikari”. Therefore, we assigned two major-effect QTLs for flowering time difference among years: one on the short arm of Chr. 11 and one on the long arm of Chr. 3. Both “Khao Nam Jen” segments increased flowering time difference. QTLs for flowering time in the corresponding years were found in both regions (Appendix A): on the long arm of Chr. 3 in all 9 years and on the short arm of Chr. 11 in 4 years.

No genes for flowering time showing DNA sequence variations between “Koshihikari” and “Khao Nam Jen” have been previously identified on the short arm of Chr. 11 (Appendix A). Two flowering-time genes—*Hd6* and *Hd16*—have been identified on the long arm of Chr. 3 and their molecular functions in flowering time regulation have been elucidated [28,29]. “Koshihikari” has non-functional alleles at these two genes, while “Khao Nam Jen” has functional alleles [30,31]. Koshihikari *Hd16*-NIL, with a functional allele in the “Koshihikari” background, had a flowering time difference of 21.6 days and CV = 5.7 (Figure 3; Appendix A). Koshihikari *Hd6*-NIL, with a functional allele in the “Koshihikari” genetic background, had a flowering time difference of 10.8 days and CV = 3.4.

### 2.3. Transcriptome Analysis in Rice Plants from Juvenile to Mature Stages

To compare gene expression patterns between “Koshihikari” and SL2812 (“Khao Nam Jen” segment on long arm of Chr. 3), we carried out RNA-seq analysis in 2015 (early-flowering year) and 2016 (late-flowering year; Figure 3). “Koshihikari” showed a transcriptome phase transition from vegetative to reproductive stages at 10 weeks after transplanting in both 2015 and 2016 (Figure 4A). SL2812 showed it at 10 weeks in 2015 and at 12 weeks in 2016 (Figure 4B). Therefore, we focused on gene expression at 10 weeks. In 2015, 95 genes were up-regulated and 234 genes were down-regulated in “Koshihikari” compared with SL2812 (Figure 4C). In 2016, 207 were up-regulated and 436 were down-regulated in “Koshihikari” (Figure 4D). Among expressed genes without significant differences in 2015, 451 genes were differentially expressed (137 up- and 314 down-regulated) in 2016 (Figure 4E). Gene ontology analysis of these 451 genes showed that they included genes involved in responses to stress stimuli, flower development and embryo development (Figure 4F,G). Moreover, expression patterns of 88 genes for phytochromes and thermo-response signaling and 59 flowering-time regulation genes fluctuated much more after 10 weeks in 2016 than in 2015 (Figure 4H,I; Appendix A). Among these 147 genes, expression of 44 genes differed more than twofold between “Koshihikari” and SL2812 after 10 weeks in 2015 and expression of 75 genes differed more than twofold in 2016. Quantitative real-time PCR confirmed different expression peaks of *Ehd1*, *Hd3a* and *RFT1*, which function downstream of *Hd16* in the flowering-time gene network, at 8 to 12 weeks in 2016 between “Koshihikari” and SL2812 (Figure 4J).

### 2.4. Effect of Weather Factors on Flowering Time in Rice Plants

To reveal weather factors associated with the difference of flowering times, we tested correlations of flowering time with air temperature, relative humidity, precipitation, wind speed, wind direction, duration of sunshine, amount of insolation and soil temperature. Maximum temperatures in July were significantly negatively correlated with flowering time in “Koshihikari”, SL2812, SL2833, Koshihikari *Hd6*-NIL and Koshihikari *Hd16*-NIL (Table 1; Appendix A). Maximum temperatures in June, August and September were significantly negatively correlated with flowering time in “Khao Nam Jen” and Koshihikari *Hd16*-NIL.

Next, we evaluated the effects of high temperature (30 °C daytime / 24 °C nighttime) and low temperature (24/20 °C) on flowering in “Koshihikari”, Koshihikari *Hd16*-NIL and SL2833. Under long daylength condition, low temperature delayed flowering in all three genotypes: the effect was 2.0 times that at normal temperature (27/24 °C) in “Koshihikari”, 2.4 times in Koshihikari *Hd16*-NIL and 2.8 times in SL2833 (Figure 5). Low temperature delayed flowering significantly in Koshihikari *Hd16*-NIL and SL2833 relative to “Koshihikari”. On the other hand, high temperature commonly led to early flowering: the effect was 0.7 times that at normal temperature in all three genotypes. Under short daylength condition, there was no significant difference in flowering time among lines, although low temperature somewhat delayed flowering time of all lines.

## 3. Discussion

### 3.1. Detection of QTLs for Response of Flowering Time to Ambient Temperature Fluctuations

Even though classical genetic studies suggest the importance of temperature response and photoperiod response in determination of flowering time in rice [21,22,23], few reports on genetic factors determine the response of flowering time to ambient temperatures. Here, we show that two major genetic factors—a novel QTL on Chr. 11 and photoperiod sensitivity gene *Hd16*—are associated with the response of flowering time to ambient temperatures during the rice-growing season.

The QTL for flowering time on the short arm of Chr. 11 was detected in only 4 out of the 9 years, with inconsistent directions of additive effects. The flowering time of SL2833, with the “Khao Nam Jen” allele at this QTL, was negatively correlated with maximum temperature in July in all 9 years; flowering was delayed in 2017 (low maximum temperature) and promoted in 2018 (high maximum temperature). Flowering time of SL2833 was significantly delayed under low temperature condition. Therefore, this QTL is associated with responsiveness to ambient temperature in regulating flowering time in rice. It was localized to the region containing *RCN1* and *RBS1* that encode phosphatidylethanolamine-binding protein and heterogeneous nuclear ribonucleoprotein R-type protein, respectively. However, the sequences of *RCN1* and *RBS1* were identical between “Khao Nam Jen” and “Koshihikari” and expression levels of both did not differ significantly in our RNA-seq dataset in 2015 and 2016. Therefore, this QTL may be a novel flowering-time gene in rice. Further genetic studies such as fine-mapping experiments are necessary to isolate the gene responsible for this QTL.

The QTL on the long arm of Chr. 3 corresponds to *Hd16*, which was isolated as a flowering time-gene involved in photoperiod sensitivity in rice [29,32,33]. SL2812 and Koshihikari-*Hd16* NIL had a wide difference of flowering time and a large CV among the 9 years. Additionally, low-temperature treatment delayed flowering time in Koshihikari-*Hd16* NIL. Our results indicate that *Hd16* is associated with both photoperiod sensitivity and response to ambient temperatures. *Hd16* encodes casein kinase-I; the “Koshihikari” allele has one non-synonymous substitution in the conserved kinase domain, which results in deficient function, causing decreased photoperiod sensitivity [29]. The functional *Hd16* allele from “Khao Nam Jen” promoted both strong photoperiod sensitivity and strong response to ambient temperatures in comparison with the non-functional “Koshihikari” *Hd16* allele. Biochemical characterization indicated that a functional Hd16 recombinant protein specifically phosphorylated the proteins encoded by rice flowering-time genes *Ghd7* and *OsPRR37* and the SLR1 protein in the gibberellin signaling pathway in rice [29,32,33]. The combined evidence suggests that *Hd16* alters the response of flowering time to ambient temperatures through both the photoperiod-sensitive flowering-time pathway and the gibberellin-dependent growth-regulatory pathway.

Several other CSSLs had moderately large flowering time fluctuations among the 9 years (Figure 2). SL2809, SL2821, SL2825 and SL2834 had more than 15 days’ difference of flowering times or CV > 5 among years. Segments from “Khao Nam Jen” were introgressed on the long arms of Chr. 2 in SL2809, Chr. 6 in SL2821, Chr. 7 in SL2825 and Chr. 11 in SL2834. Therefore, multiple genetic factors play a role in controlling flowering time under ambient temperature fluctuations in rice.

### 3.2. Integration of Thermal Response into Photoperiod Flowering Pathway

In rice, most flowering-time genes isolated previously are associated with response to daylength and photoperiod sensitivity [9,10,11,12,13]. They function in two major independent pathways, the *OsGI*–*Hd1*–*Hd3a* pathway and the *Ghd7*–*Ehd1*–*Hd3a*/*RFT1* pathway. Both pathways promote flowering under short daylength but repress it under long daylength (Appendix A). We found a novel QTL on Chr. 11 and *Hd16* on Chr. 3 that appear to control flowering time under different ambient temperatures. *Hd16* participates in the *Ghd7*–*Ehd1*–*Hd3a*/*RFT1* pathway [29,33]. Transcriptome analysis also showed fluctuations of *Ehd1*, *Hd3a* and *RFT1* expression between different ambient temperatures. Therefore, the response of flowering time to ambient temperatures would be controlled by members of the photoperiod sensitivity pathway of the *Ghd7*–*Ehd1*–*Hd3a*/*RFT1* pathway. The QTL on Chr. 11 also might function in the *Ghd7*–*Ehd1*–*Hd3a*/*RFT1* pathway, because the “Khao Nam Jen” allele did not alter the expression of *OsGI*, *Hd1*, or *Hd6* in the other pathway, or the difference of flowering time among years in the plants carrying the functional *Hd6* allele.

Extensive studies in model plant species have elucidated the details of the flowering-time genetic networks. Six major pathways have been reported in Arabidopsis: the photoperiod, vernalization, gibberellin, autonomous, age and ambient temperature pathways [34,35]. Efforts in wheat, barley, maize, sorghum, *Brassica* species and fruit trees have been also reviewed [36]. In Arabidopsis and temperate cereal crops such as wheat and barley, vernalization is a crucial factor that accelerates flowering time [15,37]. Arabidopsis has many vernalization-related genes, including *FLC*, *FRI*, *VRN1*, *VRN2*, *VIN2* and *VIN3*, but we did not detect rice homologs of these genes as QTLs in this study. Wheat and barley have three vernalization genes—*VRN1*, *VRN2* and *VRN3* (unrelated to the genes with the same names in Arabidopsis)—in each genome [38,39]. Wheat and barley *VRN1*, which encodes an AP1-like MADS-box transcription factor, is orthologous to *OsMADS14*, *OsMADS15* and *OsMADS18* in rice [40]. *VRN2*, with a zinc-finger motif and a CCT domain, seems to be homologous to rice *Ghd7* [41]. *VRN3* is homologous to *FT*-like genes such as *Hd3a* and *RFT1*. These studies suggest that homologs of rice photoperiod-sensitivity genes are associated with vernalization response in wheat and barley. Therefore, the genetic pathway that controls the response of flowering time to ambient temperatures might be conserved among cereal crop species.

High temperature accelerates plant growth and development in Arabidopsis through a signaling pathway that includes *PhyB*, *PIF4*, *YUC8*, *DELLA* and *ELF3* [16,17,18,19,20]. *PhyB* functions as a thermo-sensor in leaf epidermis and regulates *PIF4* activity and the auxin signaling pathway in response to ambient high temperature, as well as to light signals [17]. In our study, some rice orthologs of these genes showed similar expression patterns in 2015 (higher temperature year) between “Koshihikari” and SL2812, but large fluctuations in 2016 (lower temperature year) between the two. Therefore, these genes might also be associated with flowering time regulation in rice in response to ambient temperature fluctuations. Recently developed techniques in molecular biology can reveal the complex fundamental mechanisms involved in the control of the agronomically important traits [42]. To reveal the degree of conservation of the flowering-time pathway among rice, Arabidopsis and temperate cereal crops, it is necessary to identify more participant genes and to perform further molecular functional studies such as of phosphorylation, methylation and acetylation status at both gene and protein levels and to integrate those modifications into the gene pathway for flowering time regulation in rice.

### 3.3. Application to Development of Climate-Resilient Crops

Global warming is causing climate change with large temperature fluctuations, in addition to variations in solar irradiation, precipitation and soil moisture, in rice-growing regions [1,4]. Temperature fluctuations are strongly associated with crop yield losses. Novel rice cultivars resistant to both high and low temperatures are necessary for cultivation regions that are expected to experience large temperature fluctuations during the rice-growing season. In this study, we detected genetic factors for plant response to differences in ambient temperature. The two “Khao Nam Jen” alleles found change flowering time in response to ambient temperatures. If rice cultivars could promote or delay their growth and flowering time in response to ambient temperatures, we could reduce yield loss after exposure to temperature extremes at the panicle initiation, pollination and maturation stages. These “spontaneously flexible” alleles in “Khao Nam Jen” could be utilized for developing climate-resilient cultivars and thus achieving high productivity under climate change.

## 4. Conclusions

We found a wide difference of flowering time fluctuations among 9 years in Asian rice cultivars. We detected two QTLs in CSSLs with “Khao Nam Jen” chromosome segments in the “Koshihikari” genetic background. One QTL corresponds to *Hd16*, on the log arm of Chr. 3 and the other QTL, on the long arm of Chr. 11, might be a novel gene. Low temperature treatment at the panicle initiation stage delayed flowering in the two CSSLs with these QTLs. These results provide insights into thermo-sensitive flowering-time regulation in rice and will facilitate our understanding of the molecular basis of the control of flowering time variation under ambient temperature fluctuations and allow the development of climate-resilient crops that are adaptable to temperature fluctuations under climate change.

## 5. Materials and Methods

### 5.1. Plant Materials

We selected 18 rice cultivars: “Qui Zhao Zong”, “Davao 1′, “IR64′, “Touboshi”, “Deng Pao Zhai”, “Naba”, “Bei Khe”, “Bleiyo”, “Tupa 121-3′, “Kasalath”, “Muha”, “Koshihikari”, “Nipponbare”, “Basilanon”, “Khao Mac Kho”, “LAC 23′, “Khao Nam Jen” and “Nona Bokra”. These cultivars (8 *indica*, 3 *aus*, 7 *japonica*), mainly from Asia, were selected on the basis of their geographical origin, cluster analysis of genome-wide genotype data and variation in DTH from a representative rice core collection [43]. Chromosome segment substitution lines (CSSLs) with “Khao Nam Jen” chromosome segments in the “Koshihikari” genetic background were developed in our previous study [27]. The CSSLs comprise a set of 40 lines at the BC_4_F_6_ generation, covering 95.9% of the “Khao Nam Jen” genome. Two near-isogenic lines (NILs) in the “Koshihikari” genetic background were developed in previous studies [29,44].

### 5.2. Evaluation of Flowering Time in Natural Field Conditions

All plants were grown in an experimental field at the Institute of Crop Science, National Agriculture and Food Research Organization (NARO), Tsukuba, Japan (36.03° N, 140.11° E), in 9 years (2011 to 2019). Month-old seedlings were transplanted in mid-May at one per hill in plots with a double row for each line, with 18 cm between plants and 36 cm between rows. The mean daylengths were 12.3 h in April, 13.4 h in May, 14.3 h in June, 14.4 h in July, 13.6 h in August and 12.6 h in September. Cultivation management followed the standard procedures used at the Institute of Crop Science. DTH of individual plants were scored as the number of days from sowing to the appearance of the first panicle in each of the 5 or 24 plants and mean values were calculated. The difference of flowering time of each line was calculated as the difference between the earliest and latest flowering times among the 9 years. The coefficient of variation (CV) was calculated as standard deviation ÷ average flowering time in each year.

### 5.3. QTL Detection in CSSLs and NILs

The difference of flowering time and CV among years in CSSLs and NILs were compared with those of “Koshihikari” by using Dunnett”s multiple comparison procedure in JMP v. 11.0.0 software. For QTL detection in the CSSLs, we used a total of 377 SNP and 112 SSR markers showing polymorphisms between “Koshihikari” and “Khao Nam Jen” reported by the previous studies [27,30]. QTLs were declared present when these scores were significantly different between a line and “Koshihikari”. The scores were considered statistically significant if difference of flowering time >15.0 days and CV > 5.0.

### 5.4. Transcriptome at Panicle Initiation Stage

We extracted mRNA from the highest fully expanded leaf of “Koshihikari” and SL2812 (“Khao Nam Jen” segment on long arm of Chr. 3) at 2 to 20 weeks after transplanting in 2015 and 2016. Sequencing libraries for RNA-seq analysis were constructed by using the QuantSeq 3′ mRNA-seq Library Prep Kit (Lexogen, Greenland, NH, USA) with barcodes for Illumina sequencing according to the manufacturer’s instructions. The libraries were sequenced on a HiSeq 4000 platform (Illumina, San Diego, CA, USA) using the TruSeq SBS Kit (Illumina, San Diego, CA, USA) and the paired-end sequencing method to obtain two 100 bp sequences. Individual reads were trimmed by cutting bases from the start and end of reads if quality ≤20 in Trimmomatic v. 0.36 software and by removing reads with final read length <50 in FASTQC v. 0.11.9 software. Trimmed reads were mapped on the IRGSP1.0 rice genome sequence and gene loci in RAP-DB [45]. Each gene locus read was counted by featureCounts v. 1.6.4 software. Reads per million (RPM) values of all genes were calculated from the read count data. Genes with RPM ≥ 1 in all three biological replicates were defined as expressed. Expressed genes were classified according to gene-ontology types and categories from the annotations in RAP-DB [45] and RiceNETDB [46]. The RNA-seq data are archived at the DNA Data Bank of Japan under accession number DRA011161. Quantitative real-time PCR were carried out by the methods of previous study [29]. Transcription levels of *Hd3a*, *RFT1, Ehd1* and *Rice ubiquitin2* (*UBQ*) were measured according to a SYBR Green-based method by using gene-specific primers.

### 5.5. Correlation between Flowering Time and Meteorological Conditions

Weather data at the paddy field from 2011 to 2019 were collected from the Weather Data Acquisition System of the Institute for Agro-Environmental Sciences, NARO ([47], http://www.naro.affrc.go.jp/org/niaes/aws/). We collected maximum, minimum and average values of air temperature, relative humidity, precipitation, wind speed, wind direction, duration of sunshine, amount of insolation and soil temperature each month from April to November. Pearson’s correlation coefficient was used to assess relationships among DTH and the weather data in JMP.

### 5.6. Evaluation of Flowering Time under High and Low Temperatures

“Koshihikari”, Koshihikari *Hd16*-NIL and SL2833 (“Khao Nam Jen” segment on Chr. 11) were grown in a controlled-environment cabinet under a short daylength (10 h light/14 h dark) or a long daylength (14.5/9.5 h). Relative humidity was maintained at 60% under a photosynthetic photon flux density of 500 µmol m^−2^ s^−1^ provided by metal halide lamps that covered the spectrum from 300 to 1000 nm. Ambient temperatures (light / dark) were 30/24 °C, 27/24 °C (usual rice cultivation condition), or 24/20 °C. DTH in 10 plants of each line was scored and mean values were calculated for each line.

## Figures and Tables

**Figure 1 ijms-22-01024-f001:**
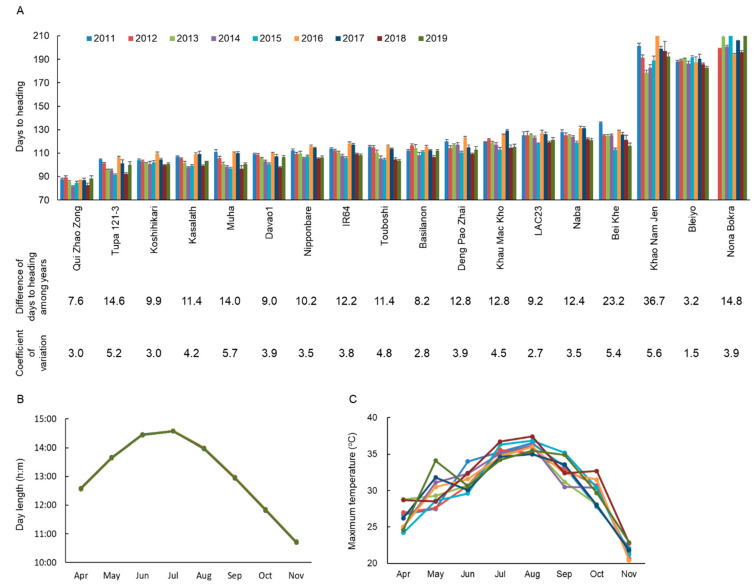
(**A**) Days-to-heading (DTH) of 18 Asian rice cultivars under natural field conditions from 2011 to 2019. Values are means ± SDs. Differences of DTH and coefficient of variation were based on the earliest and latest flowering times among years. (**B**) Daylength during rice-growing season from 2011 to 2019. (**C**) Maximum temperature during rice-growing season from 2011 to 2019.

**Figure 2 ijms-22-01024-f002:**
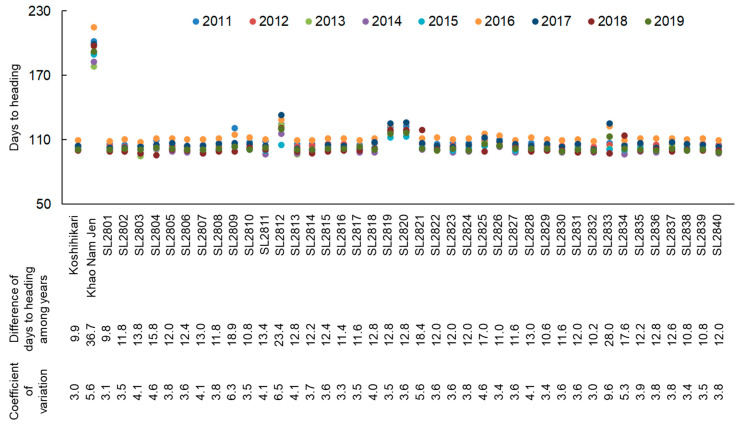
Days-to-heading (DTH) of a set of chromosome segment substitution lines (CSSLs) derived from “Koshihikari” × “Khao Nam Jen” under natural field conditions from 2011 to 2019. Differences of DTH among years were calculated from flowering times between earliest and latest years.

**Figure 3 ijms-22-01024-f003:**
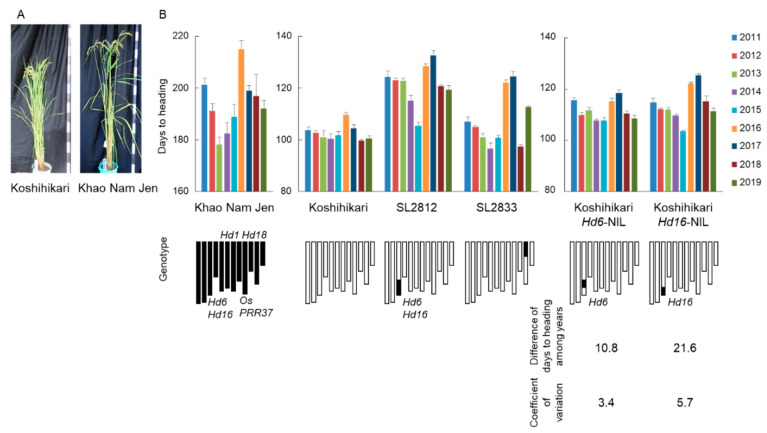
(**A**) “Koshihikari” and “Khao Nam Jen” grown under natural field conditions after flowering. (**B**) Days to heading under natural field conditions from 2011 to 2019 and its differences in “Khao Nam Jen”, “Koshihikari”, SL2812, SL2833, Koshihikari *Hd6*-NIL and Koshihikari *Hd16*-NIL.

**Figure 4 ijms-22-01024-f004:**
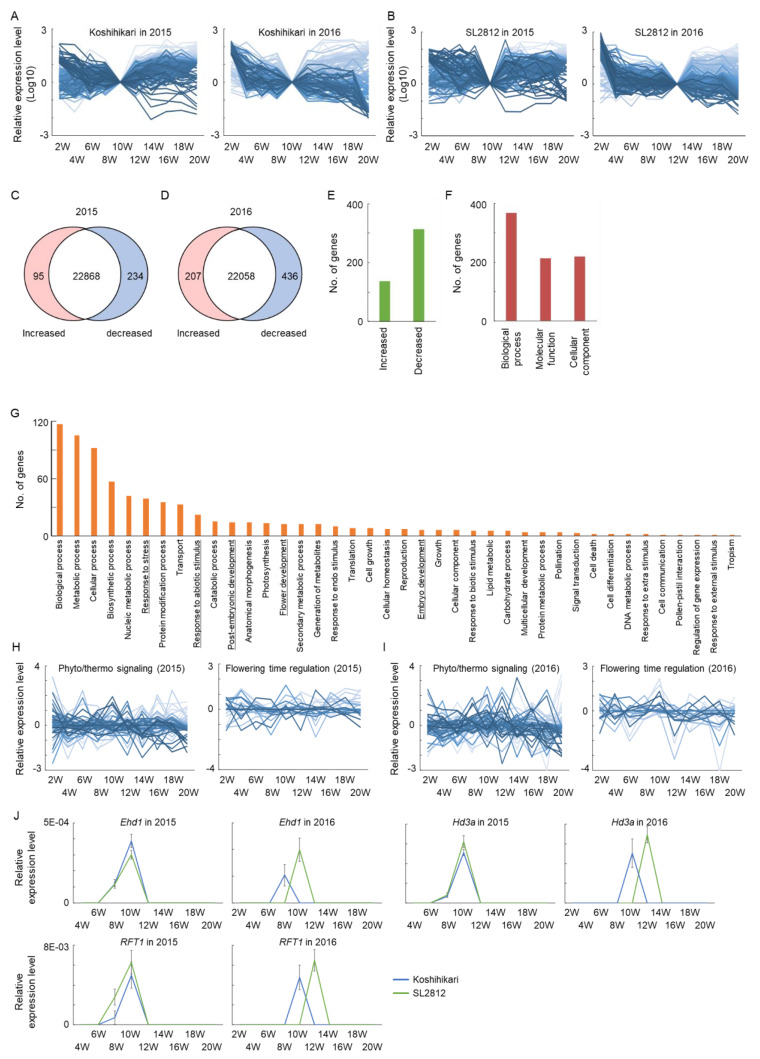
(**A**,**B**) Changes in transcriptome associated with the transition to reproductive stage in 2015 and 2016 in (**A**) “Koshihikari” and (**B**) SL2812. Relative expression patterns of 255 clusters based on 28 229 genes from 2 to 20 weeks after transplanting in the field. Dark blue lines, low expression at 20 weeks; light blue lines, high expression at 20 weeks. (**C**,**D**) Numbers of genes differentially expressed between “Koshihikari” and SL2812 at 10 weeks in (**C**) 2015 and (**D**) 2016. Increased and decreased genes were higher and lower expressions in “Koshihikari” as compared with SL2812. (**E**) Numbers of differentially expressed genes at 10 weeks in 2016 among genes showing ≤2-fold difference between “Koshihikari” and SL2812 in 2015. (**F**,**G**) Gene ontology (GO) analysis of differentially expressed genes in 2016 by (**F**) GO types and (**G**) GO categories. (**H**,**I**) Relative expression levels of phytochromes and thermo-response signaling genes and flowering time regulation genes of “Koshihikari”/SL2812 from 2 to 20 weeks in (**H**) 2015 and (**I**) 2016. (**J**) Confirmation of expression of three flowering-time genes—*Ehd1*, *Hd3a* and *RFT1*—in “Koshihikari” and SL2812 by quantitative real-time PCR.

**Figure 5 ijms-22-01024-f005:**
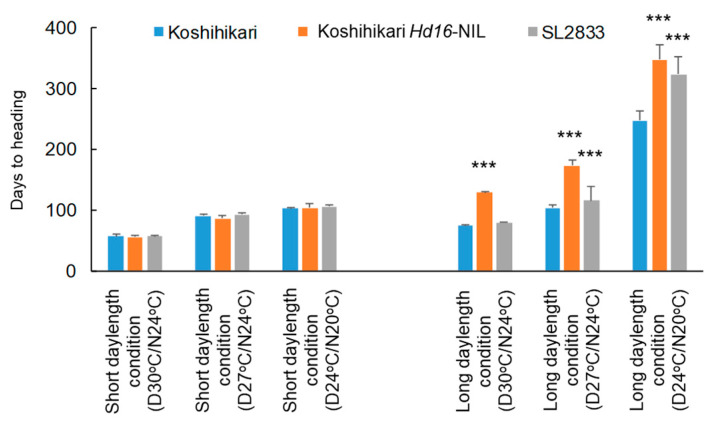
Days-to-heading of “Koshihikari”, Koshihikari *Hd16*-NIL and SL2833 under high temperature (30 °C daytime / 24 °C nighttime), normal temperature (27/24 °C) and low temperature (24/20 °C), under short daylength (10 h light/14 h dark) and long daylength (14.5 h/9.5 h). Values are means ± SDs. *** Significant difference from “Koshihikari” at *p* < 0.001 by the *t*-test.

**Table 1 ijms-22-01024-t001:** Correlations between monthly maximum temperature during rice-growing season and days-to-heading in “Koshihikari”, “Khao Nam Jen”, SL2812, SL2833, Koshihikari *Hd6*-NIL and Koshihikari *Hd16*-NIL. *** Significant correlation with days to heading at *p* < 0.001 by the Pearsons‘ test.

	Days to Heading					
	‘Koshihikari’	‘Khao Nam Jen’	SL2812	SL2833	Koshihikari *Hd6*-NIL	Koshihikari *Hd16*-NIL
Maximum temperature						
June	−0.07	0.34 ***	−0.08	−0.01	0.14	0.48 ***
July	−0.26 ***	−0.20	−0.35 ***	−0.45 ***	−0.34 ***	−0.44 ***
August	−0.16	−0.32 ***	−0.18	−0.24	−0.24	−0.42 ***
September	0.03	−0.43 ***	0.20	0.16	−0.02	−0.63 ***

## Data Availability

All datasets supporting the conclusions of this article are included in the article and Appendix A.

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
