# Peer review of "Genetic Elucidation for Response of Flowering Time to Ambient Temperatures in Asian Rice Cultivars"

_ijms, 2021, doi:10.3390/ijms22031024_

Round 1

Reviewer 1 Report

Minor comments

  1. L150-152. The sentence for qRT-PCR is unnecessary. Don’t you have the gene expression data of transcriptome?
  2. Table 1. Is there a reason why non-significant correlation to ‘Khao Nam Jen’ by July?
  3. L232-271. No figure. The reader are therefore not followed the flowering pathway.

Author Response

We greatly appreciate your taking the time to carefully consider our research and your helpful comments. And, thank you for giving us an opportunity to revise our manuscript. We revised our manuscript according to the suggestions from the reviewer. Our responses to each suggestion were written below.
We hope that with these changes, our manuscript will be acceptable for publication.

1. L150-152. The sentence for qRT-PCR is unnecessary. Don't you have the gene expression data of transcriptome?

Ans: RNA-seq analysis revealed that the three rice flowering time genes of Ehd1, Hd3a, and RFT1 differentially expressed between Koshihikari and SL2812 in 2016, but in 2015. Therefore, we confirmed these different expressions of the three rice flowering time genes by the quantitative real-time PCRs methods according to the SYBR Green-based method of previous study ([29] Shibaya et al. 2018).

2. Table 1. Is there a reason why non-significant correlation to 'Khao Nam Jen' by July?

Ans: In rice, differentiation of flower organs begins about one month before the 'days to heading, DTH' of each plant. Because DTH of the Khao Nam Jen was on October (about 190 DTH), significant correlations would be observed on August and September, not on July in the Table 1. And, because DTHs of other lines Koshihikari, SL2812, SL2833, Koshihikari Hs6-NIL and Koshihikari Hd16-NIL were on August (about 100 or 110 DTHs), significant correlations would be observed on July.

3. L232-271. No figure. The reader are therefore not followed the flowering pathway.

Ans: We showed the rice flowering time gene pathway in the Supplementary Figure S4. And, the detailed gene pathway is described by a lot of review papers published already such as [9] Shrestha et al. (2014), [10] Hori et al. (2016), [11] Brambilla et al. (2017), [12] Matsubara and Yano (2018) and [13] Liu et al. 2020.

Reviewer 2 Report

Despite the fact that rice is a short-day plant, this important food crop currently occupies a fairly wide geographical area of ​​cultivation - from areas with a temperate climate to subtropical and tropical zones with higher temperatures and different light regimes. The plasticity of rice cultivation under such differing conditions is ensured by the diversification of key genes that determine such an important trait as flowering time. The flowering time in plants, including rice, is a complex trait controlled by many quantitative trait loci (QTL). Flowering time depends on many factors, the most important of which are daylight hours and temperature. The need for research to identify genetic characteristics and molecular mechanisms underlying the formation of rice plant resistance to high temperatures is obvious. This is especially emphasized by the growing attention to the problem of global climate change on the globe. Thus, the creation of the most adapted crops to changing environmental conditions and their genetic adjustment to increase yields in conditions of various environmental problems is extremely urgent.

The authors of this manuscript used two varieties of rice, "Koshihikari" and "Khao Nam Jen", with contrasting differences in flowering time, as well as the CSSL obtained from their crossing. These samples were evaluated for flowering time over a 9 year period. As a result, two QTLs were identified, one of which is localized on Chr. 11 (detected for the first time), and the second Hd16 on Chr. 3, which takes part in controlling the flowering time at various ambient temperatures. These two QTLs found in «Khao Nam Jen» can be used to create high-yielding varieties in the face of climate change. In general, the work was carried out using modern methods of genetic and molecular analyzes, is of undoubted interest for researchers and can be recommended for publication in the journal. However, the reviewer noted two inaccuracies in the list of cited literature sources. Link [27] is indicated incorrectly, this number is not listed in the archive of the cited journal "Breeding Science". Link [44] is also given with errors: the year of publication is 2007 (2011 is the correct version) and pages 15-21 (15-22 is the correct version). The authors are recommended to carefully check the correctness of the presentation of the entire list of references.

Author Response

Ans: We greatly appreciate your taking the time to carefully consider our research and your helpful comments. And, thank you for giving us an opportunity to revise our manuscript. We revised our manuscript according to the suggestions from the reviewer. Our responses to each suggestion were written below. We hope that with these changes, our manuscript will be acceptable for publication.

We rechecked and revised all of citations in this manuscript. The reference No. [27] is published in the journal of 'Breeding Research', not in the 'Breeding Science'. And, we revised the information of the reference No. [44].

Reviewer 3 Report

Abstract: Need to be modified according to the changes in results.

Introduction: This part is fine.

Results: Need Substantial changes.

  1. Supplementary Figures are missing from the submission.
  2. Line, 82: Add oC after the mentioned temperatures 87.6 ± 1.1.
  3. Line, 86-92: These results are little confusing. What is coefficient of variation showing here? How did you decide that Bleiyo, LAC23, and Basilanon was strongly controlled by photoperiod sensitivity? It is important to explain little bit about them to make it reader friendly. However, it is understandable why did you select, Khao Nam Jen for flowering time analysis.
  4. QTL assignment is a major result of this study and Figure related to this should be in the main body not in the supplementary. There is no need of repeating the results, for example the DTH in Figure 2 and 3A. First show the QTLs here and then NILs to support your results.
  5. If there is no DNA sequence variation on Chr 11, how to explain the QTL effect?
  6. Did you applied any statistics on Figure 4J, It seems there are no significant differences in the expression.

Discussions: Modify the discussion according to changes in the results and provide sound reasoning to support your results.

Methods:

1. Explain about the methodology for QTL identifications.

Author Response

We greatly appreciate your taking the time to carefully consider our research and your helpful comments. And, thank you for giving us an opportunity to revise our manuscript. We revised our manuscript according to the suggestions from the reviewer. Our responses to each suggestion were written below.
We hope that with these changes, our manuscript will be acceptable for publication.

Results: Need Substantial changes.
1.Supplementary Figures are missing from the submission.

Ans: We resubmitted all of Supplementary Figures and Tables.

2.Line, 82: Add oC after the mentioned temperatures 87.6 ± 1.1.

Ans: The descriptions such as 87.6 ± 1.1 are days to heading (days to panicle emergence from seed sowing in the field), not temperatures. Therefore, we didn't revise these descriptions in this manuscript.

3.Line, 86-92: These results are little confusing. What is coefficient of variation showing here? How did you decide that Bleiyo, LAC23, and Basilanon was strongly controlled by photoperiod sensitivity? It is important to explain little bit about them to make it reader friendly. However, it is understandable why did you select, Khao Nam Jen for flowering time analysis.

Ans: We revised the descriptions in L. 86-L. 89. Coefficient of variation (CV) was used to evaluate dispersions of flowering time in the 18 cultivars for nine years in this study. CVs were calculated as standard deviation ÷ average flowering time in each year. And, we changed the word of 'photoperiod' sensitivity to 'daylength' in the text.

4.QTL assignment is a major result of this study and Figure related to this should be in the main body not in the supplementary. There is no need of repeating the results, for example the DTH in Figure 2 and 3A. First show the QTLs here and then NILs to support your results.

Ans: We revised Figure 3 to avoid duplicated presentations. And, Figure 2 described one of the results of QTL detections for flowering time differences (fluctuations) among the nine years. Among the CSSLs, large flowering time differences among nine years and large CVs were observed in Khao Nam Jen, SL2812 and SL2833. Supplementary Figure S2 also showed the results of QTL detection for flowering time in each year.

5.If there is no DNA sequence variation on Chr 11, how to explain the QTL effect?

Ans: As we mentioned in L. 207-L. 213, the QTL on chr 11 may encode a novel flowering-time gene, because we didn't find sequence polymorphisms in the RCN1 and RBS1 genes that were previously isolated as rice flowering time genes. To identify the causal (responsible) gene for the QTL on chr11, we need further genetic studies such as fine-mapping experiments.

6.Did you applied any statistics on Figure 4J, It seems there are no significant differences in the expression.

Ans: Figure 4J indicated differences of mRNA expression peaks for the three rice flowering time genes (Ehd1, Hd3a and RFT1) between Koshiihkari and SL2812 in 2016. As you mentioned, the mRNA expression peaks of the three genes were no significant differences in 2015. However, the mRNA expression peaks of the three genes were different between Koshihikari and SL2812 in 2016. The Ehd1 gene was highly expressioned at 8 weeks in Koshihikari, but at 10 weeks in SL2812 in 2016. The Hd3a and RFT1 genes were highly expressioned at 10 weeks in Koshihikari, but at 12 weeks in SL2812 in 2016.

Methods:

1. Explain about the methodology for QTL identifications.

Ans: We described the methods for QTL detections and their thresholds use in this study in L. 319-L. 324.

Round 2

Reviewer 3 Report

In the QTL section in Methods add or refer the the marker information it that you used for mapping.

Author Response

Ans:

L. 320-L. 322. We greatly appreciate your taking the time to carefully consider our research and your helpful comments again. We added marker information for QTL detection in this study. For the QTL detection in the CSSLs, we used a total of 377 SNP and 112 SSR markers showing polymorphisms between 'Koshihikari' and 'Khao Nam Jen' reported by the previous studies of Nagata et al. (2015) [27] and Hori et al. (2015) [30].